# Navigating Surgical Challenges: Managing Juvenile Glaucoma in a Patient with Dorfman–Chanarin Syndrome

**DOI:** 10.3390/biomedicines12102164

**Published:** 2024-09-24

**Authors:** Nicoleta Anton, Francesca Cristiana Dohotariu, Ruxandra Angela Pîrvulescu, Ileana Ramona Barac, Camelia Margareta Bogdănici

**Affiliations:** 1Department of Ophtalmology, Faculty of Medicine, Grigore T. Popa University of Medicine and Pharmacy, 700115 Iasi, Romania; camelia.bogdanici@umfiasi.ro; 2St. Spiridon Clinical Emergency Hospital, 700111 Iasi, Romania; frances7cd@gmail.com; 3Department of Ophtalmology, Carol Davila University of Medicine and Pharmacy, 050474 Bucharest, Romania; ramona.barac@umfcd.ro

**Keywords:** juvenile glaucoma, Dorfman–Chanarin Syndrome, congenital ichthyosis, strong myopia, congenital nystagmus, functional single eye, refractory glaucoma, menopause, presbyopia

## Abstract

We report a surgically challenging case, in the context of a diagnosis of juvenile glaucoma refractory to drug therapy, multi-operated, known patient with congenital ichthyosis, part of Dorfman–Chanarin Syndrome (DCS), with a single functional eye. She is a young patient (54) and housewife in an urban environment known to have DCS and BE (both eyes), strong myopia, and congenital nystagmus. She initially underwent cataract surgery in 2015 and again in 2017. As of 2015, she was known to have juvenile glaucoma under maximal therapy. The important increases in pressure started in 2020 when the dermatological condition worsened (exacerbation of skin changes in the context of ichthyosis), the patient is in menopause, and presbyopia has set in. The glaucoma could no longer be controlled with medication and required serial surgery in both eyes (initially in the right eye in 2020 and in the left eye in 2023). The right eye showed a favorable evolution until 2024, when a second trabeculectomy became necessary, with a favorable evolution. **Conclusions**: To our knowledge, such a case has not been documented in the medical literature. Frequent monitoring of intraocular pressures and prompt treatment are required. It is a rare association, a very complicated case of managing a patient with refractory glaucoma and multiple associated ophthalmic and systemic pathologies. We are also dealing with a single functional eye, difficult to manage due to a thin sclera that has caused intraoperative difficulties, and the association of congenital nystagmus and strabismus.

## 1. Introduction

Dorfman–Chanarin Syndrome is a highly rare autosomal recessive genetic disorder with systemic involvement. First described in Jerusalem in 1974 (by M.L Dorfman in 1974 and I. Chanarin in 1975), approximately 100 cases have been identified worldwide, with an equal sex ratio, and it appears in children from consanguineous couples [1,2].

The pathophysiology shows a mutation in the ABHD5 gene (chromosome 3), determining SCA fall CGI-58 (a co-factor for triacylglycerol-adipocyte lipase-ATGL), which decreases the catabolism of triglycerides and increases their accumulation in cells (skin, liver, leukocytes) [1,3,4]. Clinically, there are significant cutaneous changes (thickened, dry, parchment-like skin, progressing to desquamation, with the presence of scales over the entire skin surface), hepatic changes (hepatomegaly), muscular involvement in 60% of cases (muscle fatigue and involvement of proximal joints of the upper and lower limbs; when muscles are affected, muscle enzymes can be detected in the serum (elevated creatine kinase)40% ocular involvement (myopia, cataracts, nystagmus, strabismus), 25% hearing loss, and the same percentage is associated with intellectual impairment. Other reported manifestations include growth retardation, steatorrhea, splenomegaly, orthopedic diseases, renal dysfunction, and thyroid hypofunction [2,3,4,5,6,7]. The variability of clinical symptoms in patients with DCS depends on a large number of mutations involved. In this syndrome, liver involvement is an important cause of mortality and morbidity. Ikram Agrebi described the first reported case of DCS with renal involvement in an adult [5]. The association between DCS and renal involvement was confirmed by the presence of lipid vacuoles in the tubular epithelium. DCS, being a metabolic disease, is characterized by lipid deposits in various structures. Major clinical symptoms in patients with DCS include ichthyosis and intracytoplasmic lipid droplets [4,6,7]. DCS may present with skin changes, most commonly congenital nebular ichthyosiform erythroderma; however, erythrokeratoderma-like findings have been rarely reported in patients with DCS. Four cases of DCS presenting with different clinical forms of erythrokeratoderma have been reported: three with characteristics similar to progressive symmetric erythrokeratoderma and one resembling erythrokeratoderma variabilis. These situations are rarely associated with DCS [7,8,9,10]. Another notable case involved a 66-year-old male patient with severe cirrhosis, who required a liver transplant due to significant liver damage associated with DCS [9,10,11]. The presence of lipid vacuoles, pathognomonic for DCS, was confirmed upon evaluation. Physicians should consider DCS as a rare cause of fatty liver. The authors recommend blood smears and genetic analysis in patients presenting with severe ichthyosis, ectropion, deafness, and multiple endocrinological disorders.

The diagnosis of DCS is based on the presence of ichthyosis and identification of lipid droplets in granulocytes (Jordan’s anomaly) in a peripheral blood smear. According to the 2009 consensus, ichthyosis is divided into two groups: syndromic and non-syndromic. Dorfman–Chanarin Syndrome (DCS), a neutral lipid storage disease, was included in the classification of syndromic ichthyosis according to the consensus. In this syndrome, it was observed that regardless of the ABHD5 gene mutation, the presence of ichthyosis and lipid vacuoles (Jordan’s anomaly) in the cytoplasm of leukocytes, monocytes, and eosinophils provided differential diagnosis from other diseases. It was also noted that the coexistence of ichthyosis and Jordan’s anomaly was present in all DCS patients where the ABHD5 gene mutation was examined [5,6,10,11,12,13,14]. The paraclinical diagnosis is primarily based on genetic tests to identify the ABHD5 gene, which confirms the diagnosis [11,14,15,16]. Other investigations include liver enzyme assays (TGO/TGP) and muscle assays (CPK—Creatine phosphokinase), as well as abdominal ultrasound (hepatomegaly, hepatic steatosis). Electromyography and neurological examination are important in identifying systemic diseases. These findings broaden the clinical and mutational spectrum and emphasize the genetic heterogeneity of this disease [17,18,19,20].

Glaucoma association: There are no studies to confirm an association between this very rare syndrome and glaucoma, as in our case, so it is a unique case. The link between glaucoma and this syndrome has not been mentioned in the reviewed literature. According to glaucoma guidelines (AAO, EGS), juvenile congenital glaucomas are associated with specific genetic mutations. Juvenile glaucomas typically appear after the age of 10 and represent approximately 10% of congenital glaucomas. In open-angle glaucoma, mutations in the myocilin gene (MYOC) account for 2–4% of cases. This is why genetic testing to identify these mutations can lead to an early diagnosis and timely treatment, thereby preserving visual function. Regarding therapeutic approaches, according to the guidelines, when angle surgery is not feasible, trabeculectomy is employed, which has a success rate of 60–80%, especially when combined with Mitomycin C (MMC). Repeated failure of trabeculectomy may necessitate the use of artificial drainage systems, such as Molteno, Baerveldt, or Ahmed devices (glaucoma drainage devices), which have a success rate of 50–85% when used with intraocular pressure-lowering drops. For challenging cases that have failed multiple conservative treatments, or in cases with limited visual potential, cyclodestructive procedures are considered. These include Cyclocryotherapy, Transscleral Diode Laser Cyclophotocoagulation, Endolaser Cyclophotocoagulation, and Cycloablation/YAG for resistant cases [21,22].

## 2. Case Presentation

### 2.1. General Information about Patient

We introduce the case of a young female patient from an urban area, a housewife, with only one functional eye, who has been under the care at Saint Spiridon Ophthalmology Clinic Hospital in Iași since 2015. She has congenital ichthyosis as part of Dorfman–Chanarin Syndrome (characterized by dry, thickened skin with numerous scales on the surface that prevent the palpebral fissure from opening completely and exert pressure on the eyeballs) and advanced juvenile glaucoma in both eyes, which were operated on. She has also increased intraocular pressure in the only functional eye and significant functional and structural impairment.

### 2.2. Medical and Surgical History

RE operated for glaucoma (2020), which was decompensated with medication, and LE operated in 2023, which was compensated with medication. At presentation, the patient had the following symptomatology: red, painful RE, headache for approximately 2 weeks, known with maximal antiglaucomatous therapy AO, and was admitted for specialized investigations and treatment. She is a young patient known to have Dorfman–Chanarin Syndrome. She initially underwent cataract surgery in 2015 and again in 2017. The condition was also associated with glaucoma, which can no longer be controlled with medication and required serial glaucoma interventions in both eyes (initially in the right eye in 2020 and in the left eye in 2023). The right eye showed a favorable evolution until 2024, when a second trabeculectomy became necessary. The patient did not know the exact history of glaucoma, but only that the diagnosis was made before the cataract surgery. From the patient’s documents, in 2005, there was talk of eye damage (at the age of 38) and in 2015 (at the age of 47), glaucoma under maximum therapy was mentioned, especially congenital cataracts, strong myopia, nystagmus, and strabismus. Since 2015, she has been known to have juvenile glaucoma under maximal therapy. The important pressure increases started in 2020 when the dermatological condition worsened; the patient is in menopause, and presbyopia has set in.

Among the hereditary collateral antecedents, we mention: mother—type 2 diabetes, ischemic heart disease, osteoporosis; father—deceased (work accident); and 3 brothers—1 with psoriasis and 2 apparently healthy. The genetic consultation performed on 8 May 2009 revealed that the clinical picture suggests Dorfman–Chanarin Syndrome (the patient’s daughter is the carrier of the abnormal gene). It described at that time that the patient (38 years old) was the first child of an unrelated couple and that other cases with birth defects, genetic diseases, or reproductive disorders were NOT described in the family. The brain CT (2008) showed discrete frontal-temporal, symmetrical cortical atrophy. Audiometry revealed severe bilateral mixed hearing loss (more details about general conditions in Table 1).

### 2.3. Diagnostic Assessment and Investigations

Laboratory test: Antibody dosing: ANA; anti-LMK; antismooth muscle fiber ASMA; antimitochondrial AMA; they were normal. A recent genetic evaluation from 16 October 2023 identified a pathogenic type of the ABHD5 gene, with a phenotype suggestive of Dorfman–Chanarin Syndrome. Personal pathological antecedents showed Dorfman–Chanarin Syndrome showing congenital ichthyosis, bilateral hypoacusis (hearing aid), hypothyroidism (2011), secondary steatohepatitis (FIBROSCAN S0F3), treated essential hypertension (2011), restrictive cardiomyopathy, a minor ischemic stroke in the vertebrobasilar territory (2023), and thrombocytopenia static hypotrophy. The ophthalmologic history was as follows: BE: Advanced Juvenile Glaucoma operated RE (2020); MTD (medically decompensated for tension), LE (2023) MTC (medically compensated for tension); BE: Posterior chamber IOL RE (2015), LE (2017); BE: Pathologic myopia; BE: Orizonto-rotatory nystagmus; BE: alternating non-accommodative esotropia. She was on maximal anti-glaucoma medication for both eyes (BE). The general clinical examination revealed skin changes characterized by thickened, dry, parchment-like skin, accompanied by a sensation of tightness, with fine, white, furfuraceous scaling, and areas with polygonal, brown scales scattered across the entire skin surface, accompanied by anhidrosis at the level of the appendages, as shown in Figure 1.

The ophthalmologic examination (more details in Table 2) revealed a visual acuity in the right eye of 0.16 without correction, and in the left eye, the perception of hand movements. One eye had amblyopia due to strabismus and refractive error (high myopia). The intraocular pressure in the right eye was 24 mmHg with maximal anti-glaucoma therapy, and in the left eye, it was 18 mmHg.

Both eyes had a posterior chamber pseudophakia that was correctly positioned (initially operated for cataracts in the right eye in 2015 and then in the left eye in 2017). The filtration bleb was present, slightly prominent, encapsulated, with minimal vascularization on the surface, and had a patent peripheral iridectomy. Ocular motility revealed 30 degrees of esotropia in the right eye and 45 degrees in the left eye, with horizontal-rotatory nystagmus without a null point (Table 3 and Figure 2).

The palpebral examination revealed in both eyes, thickened, hard skin on the upper eyelids with scales on the surface, limiting the opening of the palpebral fissure. This was more pronounced in the right eye, where a lateral canthotomy was performed in 2020 (Figure 3).

The examination of the posterior segment in both eyes revealed that the fundus oculi (FOAO) was difficult to visualize due to nystagmus and the limited opening of the palpebral fissure: the optic disc was tilted with a clear border, cup-to-disc ratio (c/d) of 0.8–0.9, and large peripapillary atrophy around 360° (appearing as a myopic staphyloma), with diffuse atrophy of the retinal pigment epithelium (RPE), allowing visualization of large choroidal vessels. B-Mode Ultrasound (Eco mod B): In both eyes, the vitreous cavity was clear, the retina was attached, and the anteroposterior (AP) diameter was increased (Figure 4). Gonioscopy: In both eyes, the chamber angle was open, grade IV, with a patent site and a convex iris profile. Optical Coherence Tomography (OCT): In the right eye, there was significant thinning in all sectors of the retinal nerve fiber layers in 2024 compared to 2020. In the left eye, evaluation was not possible due to significant strabismic deviation (Figure 5).

Paraclinical investigations revealed thrombocytopenia, mild hepatocytolysis, and the absence of bacterial cultures in conjunctival secretions. Interclinical consultations revealed cardiac involvement, aortic atheromatosis, a non-dilated left ventricle (LV) with concentric hypertrophy, no segmental wall motion abnormalities, free cavities within the echocardiogram limits, no signs of pulmonary hypertension (PHT), and the absence of pericardial fluid while under medication. The dermatological consultation supported the diagnosis of congenital ichthyosis and recommended continuing treatment with emollient creams. Craniocerebral MRI scan + civ: (11 September 2023) Signal changes were present in the supratentorial, which were relatively symmetrical, and confluent white matter that could be found in cerebral ichthyosis. Based on the investigations, the diagnosis was of RE advanced juvenile glaucoma operated RE MTD (medically decompensated for tension); LE MTC (medically compensated for tension); PC-IOLBE: pathologic myopia, horizonto-rotatory nystagmus; BE: alternating non-accommodative esotropia, Dorfman–Chanarin Syndrome, congenital ichthyosis, bilateral hypoacusis, hypothyroidism, essential hypertension, and secondary steatohepatitis. From the surgical history of the right eye, it underwent a trabeculectomy in 2020, with a favorable postoperative course and an intraocular pressure (IOP) of 15 mmHg, which remained functional until the time of presentation in October 2023, with an IOP of 50 mmHg under maximal therapy. The left eye had a history of anti-glaucoma surgery, specifically a trabeculectomy in 2021, with a favorable evolution, and at the time of presentation, it had an IOP of 15 mmHg under maximal therapy. Surgery was performed on the right eye to reduce intraocular pressure and slow progression, as it was the only functional eye. Therefore, a trabeculectomy with peripheral iridectomy and antifibrotic agents (such as mitomycin C) was performed, with a favorable outcome (IOP of 14 mmHg under anti-inflammatory treatment, prominent filtration bleb, patent peripheral iridectomy, and a present anterior chamber) (Figure 6). Figure 7 illustrates the surgical and medical history of this patient (Figure 7).

### 2.4. Therapeutic Interventions and Patient Follow-Up

For four years, the anti-glaucoma surgery on the right eye kept the stability of intraocular pressure and glaucoma. However, due to subsequent decompensation of tension and the risk of disease progression, a second trabeculectomy was decided on for the only functional eye. This resulted in a very good outcome on the first postoperative day and in the following days, with a prominent and functional filtration bleb.

The subsequent evolution of this case is favorable; at the last evaluation of the right eye (one month postoperatively), visual acuity remained at 0.16 fcnc, intraocular pressure was 14 mmHg without anti-glaucoma medication, the pseudophakia was correctly positioned, and the filtration bleb was prominent, diffuse, and the peripheral iridectomy was patent, as shown in Figure 8. Due to amblyopia, the left eye maintained low visual acuity but had an intraocular pressure of 22 mmHg (under maximal anti-glaucoma therapy), with the pseudophakia correctly positioned and a prominent, functional filtration bleb.

The short-term prognosis for this patient with such pathology is favorable both generally and ophthalmologically. However, the long-term prognosis is reserved, considering the complications associated with Dorfman–Chanarin Syndrome. Ophthalmologically, the prognosis is also reserved; the long-term progression of the ophthalmologic conditions that the patient presents is uncertain, potentially leading to blindness. Social reintegration is uncertain due to having only one functional eye and multiple associated comorbidities.

## 3. Discussion

Dorfman–Chanarin Syndrome is a very rare condition associated with systemic complications and is inherited in an autosomal recessive manner, with fewer than 120 cases reported worldwide. It has an equal prevalence in males and females and occurs in children from consanguineous couples. The syndrome is caused by a mutation in the alpha/beta hydrolase domain containing 5 (ABHD5) gene, also known as comparative gene identification-58 (CGI-58), located on chromosome 3p21. It is characterized by the presence of congenital ichthyosis, decreased triglyceride catabolism, and increased accumulation in cells (skin, liver, muscles, heart, leukocytes). Lipid accumulation in various tissues occurs due to abnormal catabolism of triacylglycerols. Normally, the CGI-58 protein found on the surface of cytoplasmic lipid droplets activates lipase and leads to lipolysis. Clinically, the disease presents with congenital ichthyosis, hearing loss, hepatomegaly, splenomegaly, cirrhosis, cataracts, myopathy, and mental retardation. Skin manifestations include dryness, erythema, hyperkeratosis, and ichthyosis [17,18,19,23,24]. The differential diagnosis is always made with other syndromes that are associated with skin and ocular changes (Netherton Syndrome, Sjogren-Larsson Syndrome, Conradi-Hünermann-Happle Syndrome, FIAP Syndrome (follicular ichthyosis, atrichia, photophobia), Refsum Syndrome), which are excluded through laboratory tests and imaging. There is no curative treatment for this condition. A diet that reduces lipid intake and stops the administration of medium- and long-chain triglycerides could lower serum triglyceride (TG) levels and decrease intracellular accumulation. Another treatment method is the application of topical ointments with emollient effects. In very rare severe cases with significant liver involvement, liver transplantation is required [1,4,5,6,24,25,26]. In a review published in 2021, Erol Cakmak identified 90 articles in the literature describing Dorfman–Chanarin Syndrome (DCS): 63 were case reports, 13 were letters to the editor, 7 were original articles, 4 were short communications, and 3 were image reports. A total of 147 patients with CDS were identified, with a male predominance. Ages ranged from 0.5 to 68 years. All patients presented ichthyosis as a major symptom. Clinical examination revealed hepatomegaly in 60% of cases, bilateral ectropion in 29%, cataracts in 22%, bilateral sensorineural hearing loss in 17%, splenomegaly in 13%, strabismus in 6%, ear involvement in 7%, mental retardation in 5%, and short stature in 9% of patients identified in studies. (Table 4. highlights the clinical changes found in these 147 patients) [5].

Similar cases with this association of Dorfman–Chanarin Syndrome (DCS) are not presented in the medical literature, and there is no documentation of such a pathology in Romania. Semih Kalyon described the case of a 38-year-old patient who was found to have bilateral mixed-type hearing loss on auditory tests. Ocular involvement was bilateral, with an association of cataracts found in 46% of reported cases. In addition, nystagmus, strabismus, or ectropion may be present. This patient had cataracts, ectropion, and punctate keratopathy. In addition to general changes treated in the eyes, the patient received lubricating medication. Long-term topical therapy was administered for bilateral punctate keratopathy, with a favorable outcome [19].

The important pressure increases started in 2020 when the dermatological condition worsened, the patient was in menopause, and presbyopia had set in. A study carried out by Andrew J. Feola and collaborators found that inducing menopause in an animal model of glaucoma (ovariectomy) induced the worsening of glaucoma. The explanation is given by the role of estrogen hormones and age. The conclusions of the study support the hypothesis that menopause in younger animals would lead to greater visual impairment in OHT-exposed animals compared to older menopausal animals. hypothesis that menopause in younger animals would lead to greater visual impairment in OHT-exposed animals (ocular hypertension) and cell loss ggl compared to older menopausal animals [27].

Based on evidence that early menopause increases a female’s risk of developing glaucoma, we hypothesized that OVX (induction of menopause after ovariectomy) in young animals with OHT would result in poorer visual function than in aged animals with OHT. Contrary to this original hypothesis, the authors found that menopause caused a decline in visual function after OHT, independent of age [27].

Related to the occurrence of presbyopia and the worsening of glaucoma, there are also different hypotheses. Paul L. Kaufman et al. showed that the Accommodative Mechanism and aging are much more complex than generally believed, and extralenticular changes with age may play an important role in the pathophysiology of presbyopia, glaucomatous optic neuropathy, aqueous flow disorder, and frustrating lens incapacity current intraoculars to provide more than 1.75–2.00 diopters of dynamic accommodation, which is nowhere near sufficient for fine near vision in low light [28].

Our case presented with the association of all the changes of this syndrome (congenital ichthyosis, strabismus, nistagmus, cataract, high myopia) with juvenile glaucoma, an extremely rare association that has not been reported in the literature. Following glaucoma surgeries, the patient had a favorable outcome, with good intraocular pressure after two trabeculectomies on both eyes. The peculiarity of our case is this rare association, a very complicated case of managing a patient with treatment-refractory glaucoma and multiple associated ophthalmological and systemic pathologies. Another aspect is that we are dealing with a single functional eye, which is difficult to manage due to a thin sclera that has caused intraoperative difficulties and the association of congenital nystagmus and convergent strabismus. In our study, we are discussing a patient with a history of long-term topical anti-glaucoma medication use, who also has a dermatological condition affecting the eyelids and ocular surface, thus predisposing them to trabeculectomy failure. The guidelines also identify the following as risk factors for trabeculectomy failure: long-term use of topical medication, ocular inflammation, young age, and failure of a previous filtering surgery, even if antifibrotics (MMC) were used. The effectiveness of ophthalmic treatment is directly correlated with the patient’s systemic pathology, so a complex, multidisciplinary management approach is very important. Given the rare association of this syndrome with glaucoma, its evolution concerning intraocular pressure levels and progression is difficult to predict. The question arises as to what can be done in the event of decompensation of intraocular pressure: another surgical intervention, another trabeculectomy, an artificial drainage system, or the placement of a micro shunt in a young person with severe eyelid and ocular surface pathology. The patient requires careful monitoring of intraocular pressure and the effectiveness of the surgical intervention in the long term.

Limitations. This is a unique case, with short follow-up evolution. The absence of studies in the specialized literature does not allow the comparison of the results and its evolution regarding the level of intraocular pressure, and the progression is difficult to anticipate.

## 4. Conclusions

We introduce a highly rare case of a 54-year-old woman with Dorfman–Chanarin Syndrome (DCS) associated with glaucoma. To our knowledge, such a case has not been documented in Romania or in the medical literature. Managing such a patient is very complicated due to treatment-refractory glaucoma and multiple associated ophthalmological and systemic pathologies. Frequent monitoring of intraocular pressures and prompt treatment are necessary. The particularity of the case is numerous systemic and ophthalmologic conditions, i.e., complications from a very rare condition—Dorfman–Chanarin Syndrome and risk of trabeculectomy failure due to conjunctival adhesion formation at the filtration flap. The effectiveness of ophthalmological treatment is directly correlated with the patient’s systemic pathology, making complex, multidisciplinary management very important. Having no cases in the specialty literature to compare the evolution, the case is unique due to the difficult management and the unpredictable evolution and the presence of only one functional eye. Perhaps future studies will shed further light on this topic.

## Figures and Tables

**Figure 1 biomedicines-12-02164-f001:**
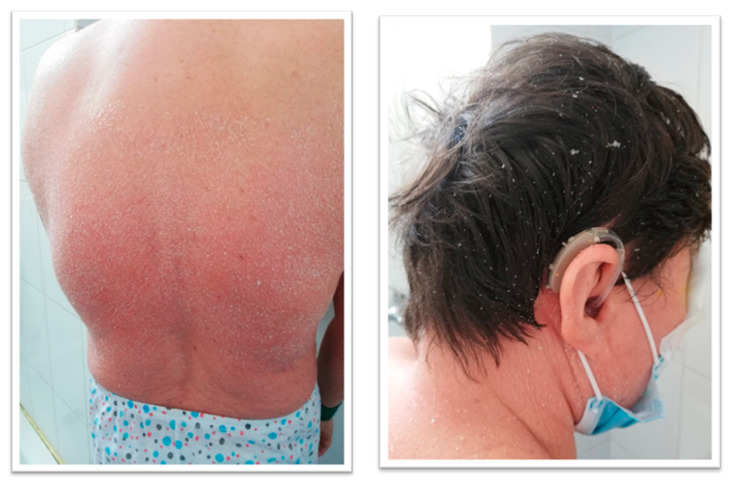
The appearance of the skin. Thickened, dry, parchment-like skin with fine, white, furfuraceous scaling (personal case study).

**Figure 2 biomedicines-12-02164-f002:**
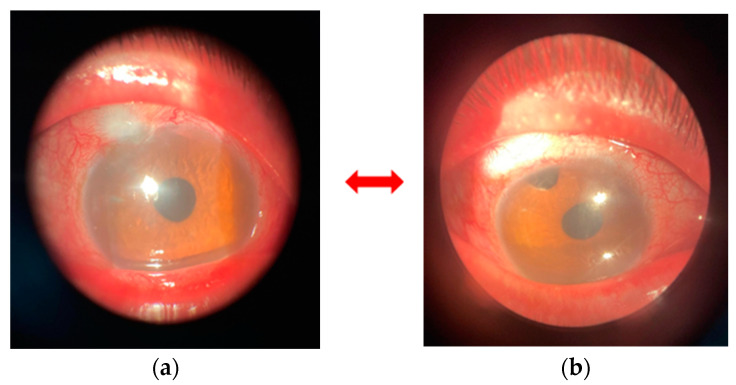
Appearance of thickened eyelids, with scales on the surface, small eyelid slit, esotropie, nistagmus orizonto-rotator. The clinicalpreoperative appearance of the filtration bleb and peripheral iridectomy in the right and left eyes shows fibrosis, particularly in the right eye (**a**,**b**).

**Figure 3 biomedicines-12-02164-f003:**
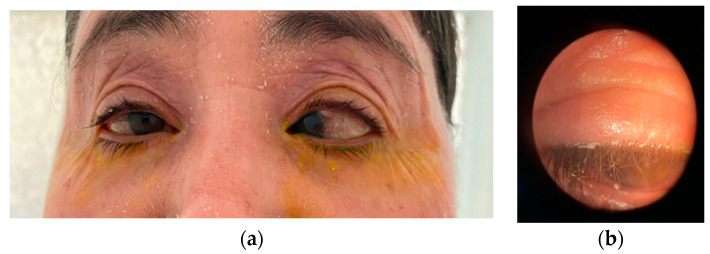
Thickened skin on the upper eyelid, scales on the surface, limited eyelid opening, narrow palpebral fissure (more pronounced in the right eye-side canthotomy 2020) (**a**,**b**) (personal case study).

**Figure 4 biomedicines-12-02164-f004:**
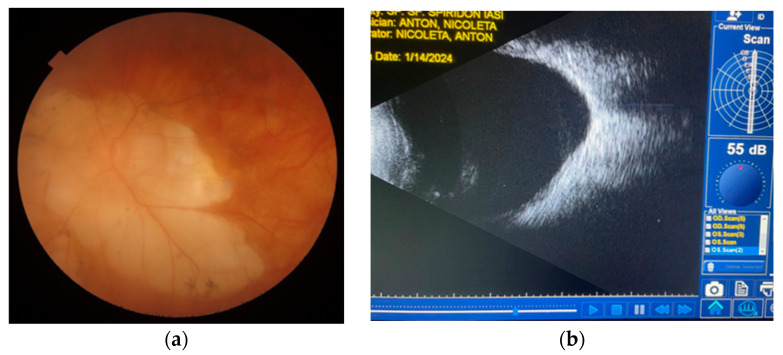
The appearance of the fundus and the ultrasound aspect revealed characteristic retinal changes of high myopia, slanted optic disc, sharp contour, c/d = 0.8–0.9, large 360° peripapillary atrophy (myopic staphyloma appearance), diffuse RPE atrophy with visualization of large choroidal vessels, (**a**) and a very long anteroposterior axis by echography (free vitreous cavity, applied retina, antero-posterior diameter) (**b**) (personal case study).

**Figure 5 biomedicines-12-02164-f005:**
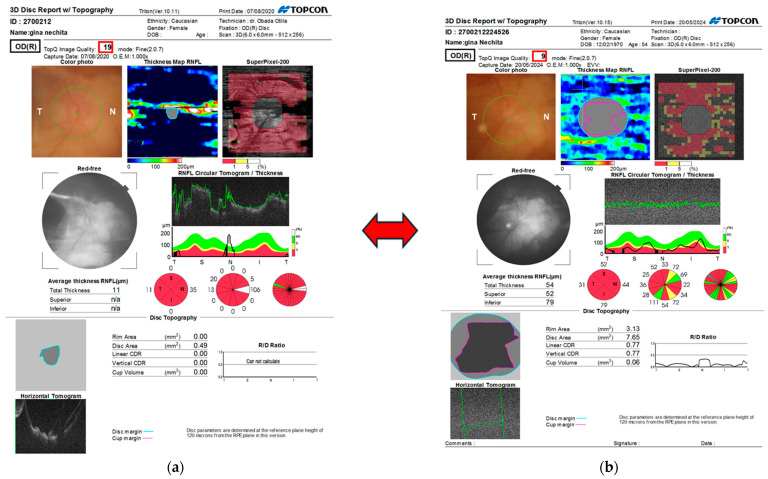
Optical Coherence Tomography (OCT) Evolution: In the right eye, there was significant thinning in all sectors of the retinal nerve fiber layer in 2024 (**b**) compared to 2020 (**a**).

**Figure 6 biomedicines-12-02164-f006:**
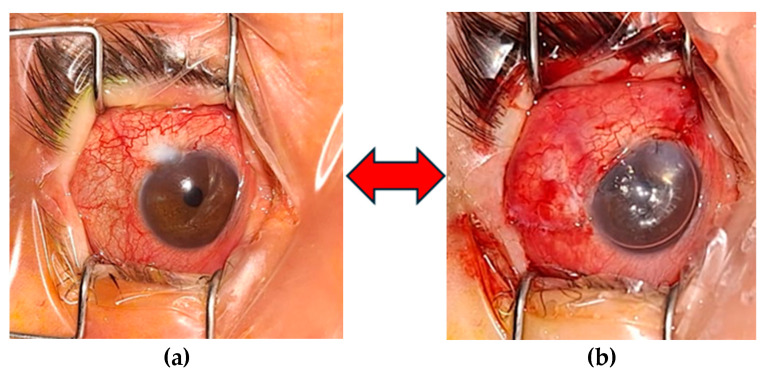
Preoperative aspect with fibrotic, cystic bleb (**a**) and postoperative aspect of the filtration bleb showing diffuse filtration (**b**) (personal case study).

**Figure 7 biomedicines-12-02164-f007:**
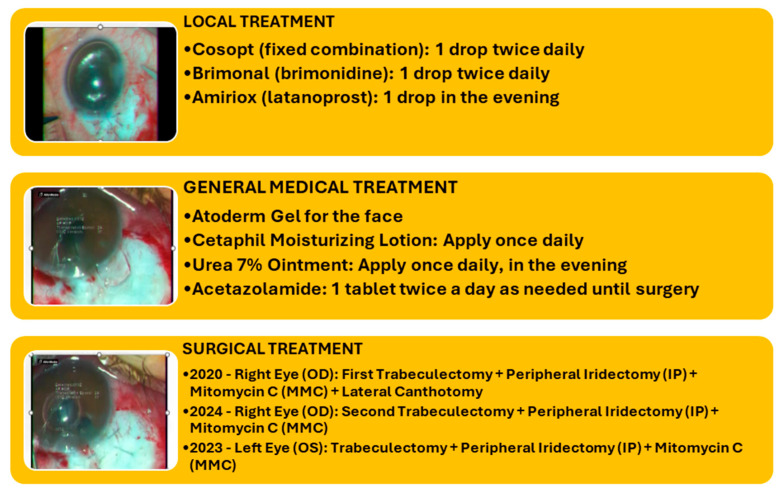
Surgical and medical history of both eyes (IP = Peripheral Iridectomy; MMC = Mitomycin C) (personal case study). The surgery was performed by the same surgeon (Nicoleta Anton), under general anesthesia, in good conditions.

**Figure 8 biomedicines-12-02164-f008:**
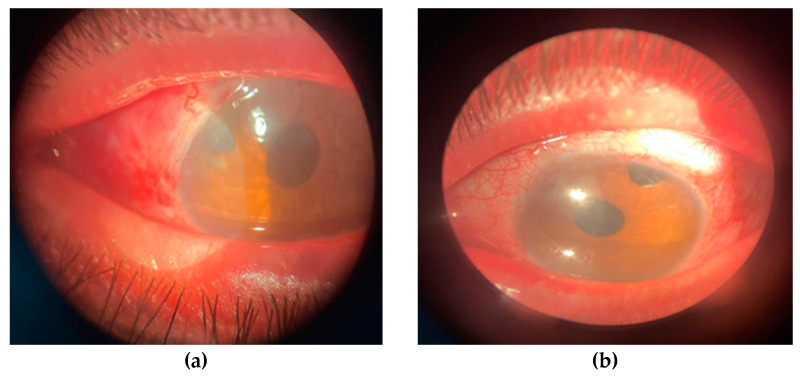
The appearance of the anterior pole 5 days after the surgery on the right eye (personal case study) (**a**) and the left eye at the time of presentation (personal case study) (**b**).

**Table 1 biomedicines-12-02164-t001:** General clinical examination.

General Condition	Slightly Influenced
Nutrition status	Static hypotrophy
State of consciousness	Present
Teguments	Thickened, dry, parchment-like skin, accompanied by a feeling of skin that feels tight, with furfuraceous scaling, white, fine, with areas of polygonal, brownish scaling scattered over the entire surface of the skin, accompanied by anhidrosis
Adipose connective tissue	Poorly represented
Limphatic ganglions	Superficial nepalpable
Muscle system	Poorly represented
Osteo-articular system	Apparent integrity

**Table 2 biomedicines-12-02164-t002:** Ophthalmological examination details.

	RE	LE
VA	0.16	perceives hand movements
Refraction	No recording due to nystagmus Ophthalmological History: Pathologic myopia	No recording due to nystagmus Ophthalmological History: Pathologic myopia
IOP	24 mmHg (under maximal antiglaucoma treatment)	18 mmHg (under maximal antiglaucoma treatment)
Chromatic sens	BE: normal	BE: normal
Light perception	BE: clear light perception in all dials	BE: clear light perception in all dials
Ocular motility	Esotropia Hirschberg 30° Orizonto-rotatory nystagmus	Esotropia Hirschberg > 45° Orizonto-rotatory nystagmus

**Table 3 biomedicines-12-02164-t003:** Biomicroscopic examination details (anterior segment pathology).

	RE	LE
Conjunctiva	Diffuse conjunctival congestion, perikeratically accentuated, upper sector FF, closed, avascularized	Mild conjunctival congestion, upper sector FF, slightly elevated, with surface vascularity pres
Sclera	Scleral thinning in the context of pathologic myopia	Scleral thinning in the context of pathologic myopia
Corneea	The cornea fixes superficially punctate fluorescein over the entire surface	The cornea fixes superficially punctate fluorescein over the entire surface
Anterior chamber	Deep, free	Deep, free
Iris	Peripheral iridectomy at 11 a.m., areas of iridal atrophy in the Temporal, Nazal sector	Peripheral iridectomy at 1 p.m., areas of iridal atrophy in sector Temporal , Nazal
Pupil	Round, central, weak reflex	Round, central, weak reflex
Lens	PC-IOL transparent, correctly positioned	PC-IOL transparent, correctly positioned
Orbits	Orbital rim integral	Orbital rim integral
Eyelid	BE: Thickened, hard upper eyelid teguments with scales on the surface limiting the opening of the palpebral fissure. (More pronounced in the right eye)	BE: Thickened, hard upper eyelid teguments with scales on the surface limiting the opening of the palpebral fissure. (More pronounced in the right eye)
Lacrimal sistem	BE: Tear hyposecretion, severe dry eye syndrome	BE: Tear hyposecretion, severe dry eye syndrome

**Table 4 biomedicines-12-02164-t004:** Summary of patient’s clinical data (5).

Clinical Features	Procente (%)
hepatomegaly	88 (60)
bilateral ectropion	42 (29)
neurosensory deafness	25 (17)
splenomegaly	19 (13)
strabismus	9 (6)
small ear	11 (7)
mental retardation	8 (5)
short posture	13 (9)
myopathy	86 (59)

## Data Availability

The datasets used and analyzed in this study are available from the corresponding author on reasonable request.

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
