# Peer review of "Navigating Surgical Challenges: Managing Juvenile Glaucoma in a Patient with Dorfman–Chanarin Syndrome"

_biomedicines, 2024, doi:10.3390/biomedicines12102164_

Round 1

Reviewer 1 Report

Comments and Suggestions for Authors

See the attachment

Comments on the Quality of English Language

See the attachment for authors

Author Response

Reviwer 1.

Dear reviewer,

We highly appreciate your thoughtful comments and we hope that, with the changes carried out, our paper Surgical Challenges in the treatment of glaucoma in a patient with Dorfman-Chanarin Syndrome , Biomedicines NR 3167410

We have improved the quality of the article according to what you suggested, for which we thank you, as well for your professional evaluation, time and efort! Your suggestions improved our manuscript. In case of any other suggestions, we will try to address it immediately in solving it.

The detailed point-by-point responses to your comments are given below

The abstract and introduction describe the main features of this rare genetic condition, DorfmanChanarin syndrome (SCD), characterised by systemic, ocular (mainly refractory juvenile glaucoma and cataracts) and laboratory and haematochemical alterations. The article is  structured in such a  way that first a description of SCD is given and then a clinical case of a patient with SCD is reported. In the latter section, particular emphasis is placed on medical but especially surgical therapies, making it clear that further studies are needed in this area given the complexity of this syndrome. Furthermore, it could be useful in this section to implement the bibliography with more recent citations. For instance:

  • Four cases of Chanarin-Dorfman syndrome presenting with different types of

erythrokeratoderma,Tubanur Çetinarslan et al. Pediatr Dermatol. 2024

  • Chanarin-Dorfman Syndrome diagnosed at the stage of liver transplantation: A rare lipid

storage disease. Esra Durmazer et al. J Clin Lipidol. 2024

  • Dorfman-Chanarin Syndrome with Renal Involvement: A Rare Case Report and Literature

Review, Ikram Agrebi et al. Indian J Nephrol. 2023 Nov-Dec.

Returning to the introduction, it appears to be well written and well elaborated, the correlation

between SCD and the main genetic mutation, ABHD5, is emphasised, and the main clinical

manifestations, i.e. hepatic, muscular, ocular and cutaneous, are described. Furthermore, the

article points out that, although SCD patients very often also suffer from glaucoma, there are no

studies in the literature that clearly demonstrate this correlation, thus highlighting the need for

this study and future studies on this topic, where our knowledge is still lacking.

The clinical case describes a 54-year-old patient, suffering from SCD, with a particularly complex

medical history (ichthyosis, horizontal-rotatory nystagmus, severe myopia, skin manifestations,

etc.).

In the next section, the clinical evolution of the patient over the following 4 years is described. The

study shows how the IOP has been stabilised, although further intervention is probably necessary,

and how the dermatological manifestations have evolved over time. The iconographic part helps

and facilitates comprehension, the images are of good quality, relevant and pertinent to what is

written while the tables are simple but well structured.

Finally, the discussion highlights and underlines the complexity of this syndrome, given by the

intersection of several systemic and non-systemic pathologies. It emphasises both the need for

further studies to better understand the pathophysiology of SCD and the correlation with

glaucoma and the need for a true multidisciplinary and personalised approach to the patient.

In conclusion, the study delves into a rare syndrome, enriches the literature on a subject where we

have yet to discover much, describes diagnostic and therapeutic methods performed, and

provides a useful reference for similar clinical cases in the future. The authors suggest that further

research is needed to improve therapeutic strategies and provide clearer guidelines for the

treatment of patients with similar conditions. Obviously, as this is a case report, it is difficult to

generalise the results obtained, but the article is certainly a starting point and provides valuable

insights into surgical treatment. In conclusion, the article is well written and well structured, the

work is scientifically sound, however it could benefit from some improvements:

  • A more detailed section on glaucoma, if possible listing more surgical, alternative

and/or complementary treatment options.

Response

  • Citing more recent scientific articles dealing with SCD in the introductory section.

  • Inclusion of international guidelines such as AAOO and EGS.

Response

Indeed, there are recent case studies on this rare syndrome. I wanted to highlight the studies that associate more ophthalmological pathology with this dermatological pathology. But I studied them and cited them because they add to the work by highlighting the multiple rare situations encountered and the way to solve them., marked in yellow in the text and bibliography. This case of ours is particularly associated with glaucoma and the difficulty in surgical management due to the associated ocular surface disease.

Ikram Agrebi1 describes the first case of DCS with renal involvement reported in an adult. The association between the two was confirmed by the presence of lipid vacuoles in the tubular epithelium. Being a metabolic disease, DCS consists of lipid deposits in several structures. Major clinical symptoms in patients with CDS include ichthyosis and intracytoplasmic lipid droplets. CDS may present with skin changes, most commonly congenital nebular ichthyosiform erythroderma, however findings similar to erythrokeratoderma have been rarely reported in patients with CDS. Four patients with CDS presenting different clinical forms of erythrokeratoderma are reported (three with characteristics similar to progressive symmetric erythrokeratoderma and one with characteristics similar to erythrokeratoderma variabilis), situations rarely reported in association with DCS. (Tubadur et al., 2024). Another particular situation of association of DCS with significant liver damage that required a liver transplant (3Esra Durmazer, 2024), patient of A 66-year-old male patient with severe cirrhosis and who, following the evaluations, demonstrated the presence of lipid vacuoles pathognomonic for DCS. Physicians should be aware of CDS as a rare cause of fatty liver. The authors recommend a blood smear and genetic analysis in patients with severe ichthyosis, ectropion, deafness and multiple endocrinological disorders.

The association of glaucoma with this syndrome is not cited in the specialized literature. According to the glaucoma guidelines (AAO, EGS), juvenile congenital glaucomas are associated with specific genetic mutations. Juvenile glaucomas appear later, after the age of 10 and are a percentage of 10% of congenital glaucomas.  In open-angle glaucoma, we are talking about mutations of the myocilin gene (MYOC) IN 2-4% of glaucomas. That is why genetic testing with the identification of this gene can determine an early diagnosis and early adequate treatment, with the preservation of visual function. When we talk about therapeutic behavior, according to the guidelines, when angle surgery is not possible, trabeculectomy is used, which has a success rate between 60 and 80% when it is also associated with MMC (mitomycin C). The repeated failure of trabeculectomy leads to the use of artificial drainage systems (such as Molteno, Baerveldt, and Ahmed device implantation (glaucoma drainage devices), with a 50%–85% success rate with IOP-lowering drops. Treatments for challenging cases that have failed multiple, more conservative treatments, and cases with limited visual potential: cyclodestructive procedures such as Cyclocryotherapy, Transscleral diode laser cyclophotocoagulation, Endolaser cyclophotocoagulation and Cycloablation/YAG for resistant cases.

In the case of our study, we are discussing a patient with a history of long-term topical anti-glaucomatous medication, with a dermatological disease that involves the eyelids and the ocular surface, thus predisposing to trabeculectomy failure. The guidelines also discuss as risk factors of tracheulectomy failure: topical medication administered for a long time, eye swelling, young age and failure of a previous filtering surgery even if anthofibrotics (MMC) were used.

She is a young patient known to have dorfmann-chanarin syndrome, initially operated on for cataracts in 2015 and then in 2017 and associated with glaucoma that can no longer be compensated with medication and requires serial glaucoma interventions in both eyes (2020 initially at RE and 2023 at LE) . With a favorable evolution at RE until 2024 when he needs a second trabeculectomy.  We are discussing a patient with a history of long-term topical antiglaucoma medication, with a dermatological blah involving the eyelids and ocular surface, thus predisposing to the failure of trabeculectomy. Having no cases in the specialty literature to compare the evolution, the case is unique due to the difficult management and the unpredictable evolution and the presence of only one functional eye.

As well as treatment directions according to the AAO page 104 and EGS guidelines. In the case of our patient Trabeculectomy failure or risk of failure, EGS recommends artificial drainage systems such as Ahmed, Molteno

Reviewer 2 Report

Comments and Suggestions for Authors

The authors of this study present a case of a 54-year-old with juvenile glaucoma currently decompensated, combined with Dorfman-Chanarin syndrome-related congenital ichthyosis, myopia, congenital nystagmus, and postmenopausal hormonal imbalance. The topic selected for this study is innovative, but the following problems could be improved:

1.      Please check that spelling and abbreviations are correct in the article and ensure that each abbreviation is followed by a detailed explanation of the full name (e.g., CDS does not have a corresponding explanation), to avoid inconsistencies in terminology and errors in grammar and word usage.

2.      In the introduction, it is mentioned that there are no studies demonstrating an association between DCS and glaucoma (page 2, line 48-49). However, the authors' lack of exploration of potential links or hypotheses between the two diseases, whether the association is coincidental or whether there may be an underlying mechanism linking the two, requires further clarification of the purpose of the study in relation to the potential link between the two.

3.      The authors list the history of glaucoma surgery and medications performed on this patient (page 2, line 75). Based on the patient's multiple chronic ophthalmic and systemic comorbidities, all pharmacological and surgical treatments, if any, should be listed in the text (including cataract surgery and laser therapy, which are not mentioned) to avoid other medications that may predispose to, or influence, the prognosis and development of glaucoma.

4.      While ensuring the quality of the pictures, do not repeat the use of pictures in the text (e.g., Figures 2 and 3), note that each picture should have a legend and an introduction (e.g., Figure 5), the pictures should be focused, and more than one picture should be labelled with the serial number a, b. It is recommended that they be deleted if they do not explain the content of the article in a good and in-depth manner.

5.      In the results section the authors focus on the use of surgical interventions to control intraocular pressure (IOP) in the patient's only normal eye (page 6, line 78). However, it is not clear whether the change in the patient's glaucomatous condition was related to the choice of prior surgery or the development of the patient's own multiple systemic diseases, or to hormonal changes during menopause. Changes in other disease indicators in glaucomatous loss need to be described to establish a link between the diseases.

6.      Given that the authors report a rare association between DCS and juvenile glaucoma, the discussion should be more reflective of the evidence that there is a link between the two diseases, or that juvenile glaucoma is associated with altered genetic material. More detailed discussion and reflection is needed regarding long-term treatment strategies for patients.

Author Response

Reviwer 2

Dear reviewer,

We highly appreciate your thoughtful comments and we hope that, with the changes carried out, our paper Surgical Challenges in the treatment of glaucoma in a patient with Dorfman-Chanarin Syndrome , Biomedicines NR 3167410

We have improved the quality of the article according to what you suggested, for which we thank you, as well for your professional evaluation, time and efort! Your suggestions improved our manuscript. In case of any other suggestions, we will try to address it immediately in solving it.

The detailed point-by-point responses to your comments are given below

The authors of this study present a case of a 54-year-old with juvenile glaucoma currently decompensated, combined with Dorfman-Chanarin syndrome-related congenital ichthyosis, myopia, congenital nystagmus, and postmenopausal hormonal imbalance. The topic selected for this study is innovative, but the following problems could be improved:

Feedback

  1. Please check that spelling and abbreviations are correct in the article and ensure that each abbreviation is followed by a detailed explanation of the full name (e.g., CDS does not have a corresponding explanation), to avoid inconsistencies in terminology and errors in grammar and word usage.

Response

I made the changes

  1. In the introduction, it is mentioned that there are no studies demonstrating an association between DCS and glaucoma (page 2, line 48-49). However, the authors' lack of exploration of potential links or hypotheses between the two diseases, whether the association is coincidental or whether there may be an underlying mechanism linking the two, requires further clarification of the purpose of the study in relation to the potential link between the two.

I made the changes. I added a paragraph marked in yellow

  1. The authors list the history of glaucoma surgery and medications performed on this patient (page 2, line 75). Based on the patient's multiple chronic ophthalmic and systemic comorbidities, all pharmacological and surgical treatments, if any, should be listed in the text (including cataract surgery and laser therapy, which are not mentioned) to avoid other medications that may predispose to, or influence, the prognosis and development of glaucoma.

I made the changes. I added a paragraph marked in yellow.

  1. While ensuring the quality of the pictures, do not repeat the use of pictures in the text (e.g., Figures 2 and 3), note that each picture should have a legend and an introduction (e.g., Figure 5), the pictures should be focused, and more than one picture should be labelled with the serial number a, b. It is recommended that they be deleted if they do not explain the content of the article in a good and in-depth manner.

I made the changes.

  1. In the results section the authors focus on the use of surgical interventions to control intraocular pressure (IOP) in the patient's only normal eye (page 6, line 78). However, it is not clear whether the change in the patient's glaucomatous condition was related to the choice of prior surgery or the development of the patient's own multiple systemic diseases, or to hormonal changes during menopause. Changes in other disease indicators in glaucomatous loss need to be described to establish a link between the diseases.

Este o pacienta tanara cunoscuta cu sindrom dorfmann chanarin, operata initial de cataracta in 2015 si apoi in 2017 si care asociaza glaucom ce nu mai poate fi compensat medicamentos si necesita iterventii seriate de glaucom la ambii ochi (2020 initial la RE si 2023 la LE). Cu o evolutie favorabila la RE pana in 2024 cand necesita o a doua trabeculectomie.  Discutam de o pacienta ce are istoric de medicatie antiglaucomatoasa topica indelungata, cu o boala dermatologica ce implica pe langa pleoape si suprafata oculara, predispozand astfel la esecul trabeculectomiei. Neavand cazuri in literatura de speciialitate pentru compararea evolutiei, cazul este unic prin managemantul dificil si evolutia imprevizibila si prezentei unui singur ochi functional.

Ca si directii de tratament conform ghidurilor AAO pagina 104 si EGS. In cazul pacientei noastre Esecul trabeculectomiei sau riisc de esec , EGS recomanda sisteme artificiale de drenaj de tipul Ahmed, Molteno

  1. Given that the authors report a rare association between DCS and juvenile glaucoma, the discussion should be more reflective of the evidence that there is a link between the two diseases, or that juvenile glaucoma is associated with altered genetic material. More detailed discussion and reflection is needed regarding long-term treatment strategies for patients.

Glaucoma association. There are no studies confirming an association between this very rare syndrome and glaucoma. The link between glaucoma and this syndrome is not mentioned in the reviewed literature. According to glaucoma guidelines (AAO, EGS), juvenile congenital glaucomas are associated with specific genetic mutations. Juvenile glaucomas typically appear after the age of 10 and represent approximately 10% of congenital glaucomas. In open-angle glaucoma, mutations in the myocilin gene (MYOC) account for 2-4% of cases. This is why genetic testing to identify these mutations can lead to an early diagnosis and timely treatment, thereby preserving visual function. Regarding therapeutic approaches, according to the guidelines, when angle surgery is not feasible, trabeculectomy is employed, which has a success rate of 60-80%, especially when combined with Mitomycin C (MMC). Repeated failure of trabeculectomy may necessitate the use of artificial drainage systems, such as Molteno, Baerveldt, or Ahmed devices (glaucoma drainage devices), which have a success rate of 50-85% when used with intraocular pressure-lowering drops. For challenging cases that have failed mul-tiple conservative treatments, or in cases with limited visual potential, cyclodestructive procedures are considered. These include Cyclocryotherapy, Transscleral Diode Laser Cyclophotocoagulation, Endolaser Cyclophotocoagulation, and Cycloablation/YAG for resistant cases.

I have added in yellow a paragraph about the association of glaucoma to the introduction and to the discussions

Round 2

Reviewer 2 Report

Comments and Suggestions for Authors

The authors have made significant improvements to the manuscript based on my initial review. There are still some areas that require further clarification and elaboration to ensure the manuscript meets the standards for publication:

The manuscript would benefit from a more comprehensive account of the patient’s glaucoma history. A thorough analysis of the relationship between prior cataract surgery and the subsequent development of glaucoma should be included to provide a more complete clinical picture. Additionally, the discussion section requires a deeper exploration of the uncommon association between DCS and glaucoma. It would be advantageous to present evidence linking these two conditions or to integrate considerations regarding juvenile glaucoma in the context of genetic alterations, supported by relevant literature.

Author Response

Rouds 2

Reviewer

The authors have made significant improvements to the manuscript based on my initial review. There are still some areas that require further clarification and elaboration to ensure the manuscript meets the standards for publication:

The manuscript would benefit from a more comprehensive account of the patient’s glaucoma history. A thorough analysis of the relationship between prior cataract surgery and the subsequent development of glaucoma should be included to provide a more complete clinical picture. Additionally, the discussion section requires a deeper exploration of the uncommon association between DCS and glaucoma. It would be advantageous to present evidence linking these two conditions or to integrate considerations regarding juvenile glaucoma in the context of genetic alterations, supported by relevant literature.

Response

Dear reviewer,

We highly appreciate your thoughtful comments and we hope that, with the changes carried out, our paper Surgical Challenges in the treatment of glaucoma in a patient with Dorfman-Chanarin Syndrome , Biomedicines NR 3167410

We have improved the quality of the article according to what you suggested, for which we thank you, as well for your professional evaluation, time and efort! Your suggestions improved our manuscript. In case of any other suggestions, we will try to address it immediately in solving it.

The detailed point-by-point responses to your comments are given below

In response to the first part.The patient does not know the exact history of glaucoma, but only that the diagnosis was made before the cataract surgery. From the patient's documents, in 2005 there is talk of eye damage (at the age of 38) and in 2015 (at the age of 47, it is mentioned about glaucoma under maximum therapy), especially congenital cataracts, strong myopia, nystagmus, strabismus. Since 2015, she has been known to have juvenile glaucoma under maximal therapy. The important pressure increases started in 2020 when the dermatological condition worsened, the patient is in menopause and presbyopia has set in.

A study carried out by Andrew J. Feola and collaborators found that inducing menopause in an animal model of glaucoma (ovariectomy induces the worsening of glaucoma. The explanation is given by the role of estrogen hormones and age. The conclusions of the study support the hypothesis that menopause in younger animals would lead to greater visual impairment in OHT-exposed animals compared to older menopausal animals. hypothesis that menopause in younger animals would lead to greater visual impairment in OHT-exposed animals (ocular hypertension) and cell loss ggl compared to older menopausal animals.

Based on evidence that early menopause increases a female's risk of developing glaucoma, we hypothesized that OVX (induction of menopause after overectomy) in young animals with OHT would result in poorer visual function than in aged animals with OHT. Contrary to this original hypothesis, the authors found that menopause caused a decline in visual function after OHT, independent of age.

Related to the occurrence of presbyopia and the worsening of glaucoma, there are also different hypotheses. Paul L. Kaufman et al., show that the Accommodative Mechanism and aging are much more complex than generally believed, and extralenticular changes with age may play an important role in the pathophysiology of presbyopia, glaucomatous optic neuropathy, aqueous flow disorder, and frustrating lens incapacity current intraoculars to provide more than 1.75 to 2.00 diopters of dynamic accommodation, which is nowhere near sufficient for fine near vision in low light.

This association of Dorfman-Chanarin Syndrome (DCS)and glaucoma are not presented in the medical literature. I have not found studies that provide additional information regarding the causal relationship between this genetic syndrome and the occurrence and worsening of glaucoma. Perhaps future studies will bring additional clarification vis vis vis this subject. If you have any information, I would be very pleased to find it out.

Thank you so much for your review!